# Optimal dose of perineural dexamethasone for the prolongation of analgesia for peripheral nerve blocks: protocol for a systematic review and meta-analysis

Wenjie Chai [1], Shaopeng Wang,[1] Donghang Zhang[2]

¹Department of Anesthesiology, Lanzhou Traditional Chinese Medicine Hospital of Orthopedics and Traumatology, Lanzhou, Gansu, China
²Department of Anesthesiology, Sichuan University West China Hospital, Chengdu, Sichuan, China

**Correspondence to**
Dr Donghang Zhang;
zhangdhscu@163.com and
Dr Wenjie Chai;
119251554@qq.com

## ABSTRACT

**Introduction** Perineural use of dexamethasone is demonstrated to extend the analgesia duration of peripheral nerve blocks (PNB), but its optimal dose remains unclear. This systematic review and meta-analysis aims to determine the optimal dose of perineural dexamethasone in the prolongation of analgesia for PNB.
**Methods and analysis** PubMed, EMBASE, the Cochrane Central Register of Controlled Trials and Web of Science will be searched from their inception to 1 March 2023. Language will be restricted to English. Randomised controlled trials that compared the efficacy and safety of different doses of perineural dexamethasone for PNB in adult patients will be included. Retrospective studies, reviews, meta-analyses, case reports, conference abstracts, comments and studies regarding paediatric surgeries will be excluded. The duration of analgesia will be defined as the primary outcome. Secondary outcomes will include pain scores, the total analgesic requirement over 48 hours and the incidence of adverse effects. Two reviewers will independently perform the study selection, data extraction and quality assessment. RevMan V.5.3 software will be used for data analysis. The quality of evidence will be assessed using the Grading of Recommendation, Assessment, Development and Evaluation (GRADE) approach.
**Ethics and dissemination** No ethical approval is required. The results of this study will be submitted to peer-reviewed journals.
**PROSPERO registration number** CRD42022385672.

## STRENGTHS AND LIMITATIONS OF THIS STUDY

⇒ The Grading of Recommendation, Assessment, Development and Evaluation (GRADE) approach will be used to assess the quality of evidence.
⇒ Subgroup analyses, meta-regression and sensitivity analyses will be performed to explore heterogeneity, and to attempt to identify the optimal dose of perineural dexamethasone for a specific type of peripheral nerve block.
⇒ High clinical heterogeneity may exist due to several factors, such as the type and concentration of local anaesthetics, type of surgery, anaesthesia and nerve blocks.

## INTRODUCTION

Peripheral nerve blocks (PNB) are widely used for anaesthesia and/or analgesia; however, the analgesic effects of single-shot PNB last only for a few hours.[1] Several adjuvants have been used to prolong the analgesic duration of PNB, including intravenous or perineural injection of dexamethasone, dexmedetomidine and opioids.[2 3] Dexamethasone is the most commonly used adjuvant for PNB.[4 5] Perineural dexamethasone is used more commonly for PNB than the intravenous route, and a previous meta-analysis suggested that perineural dexamethasone provided longer duration of analgesia compared with intravenous dexamethasone when used as an adjuvant in PNB.[4 6] Currently, many studies have compared the effects of various doses of perineural dexamethasone on PNB, such as 1 mg, 2 mg, 4 mg, 5 mg and 8 mg,[7–20] but the results remain controversial. Two previous meta-analyses[21 22] have evaluated the effects of different doses of perineural dexamethasone on PNB. One, published in 2015, showed no differences in the safety and efficacy between 4 and 8 mg perineural dexamethasone for PNB.[22] Another, published in 2018, reported that 4 mg of perineural dexamethasone represented a ceiling dose for the prolongation of analgesic duration when used in combination with local anaesthetics for PNB.[21] Recently, several new randomised controlled trials (RCTs) comparing the effects of different doses of perineural dexamethasone on PNB have been reported, but yielded inconsistent results.[12 13 16 17 20 23] Therefore, it is appropriate to conduct an updated systematic review and meta-analysis to provide the latest evidence

**Table 1** Search strategy for PubMed

| | Search terms |
|---|---|
| #1 | Dexamethasone [Mesh] |
| #2 | Dexamethasone [Title/Abstract] |
| #3 | #1 OR #2 |
| #4 | Nerve block [Title/Abstract] |
| #5 | Nerve blockade [Title/Abstract] |
| #6 | Peripheral nerve blockade [Title/Abstract] |
| #7 | Peripheral nerve block [Title/Abstract] |
| #8 | #4 OR #5 OR #6 OR #7 |
| #9 | Randomized controlled trial [Title/Abstract] |
| #10 | Clinical study [Title/Abstract] |
| #11 | Controlled clinical trial [Title/Abstract] |
| #12 | Clinical trial [Title/Abstract] |
| #13 | Randomized [Title/Abstract] |
| #14 | #9 OR #10 OR #11 OR #12 OR #13 |
| #15 | #3 AND #8 AND #14 |

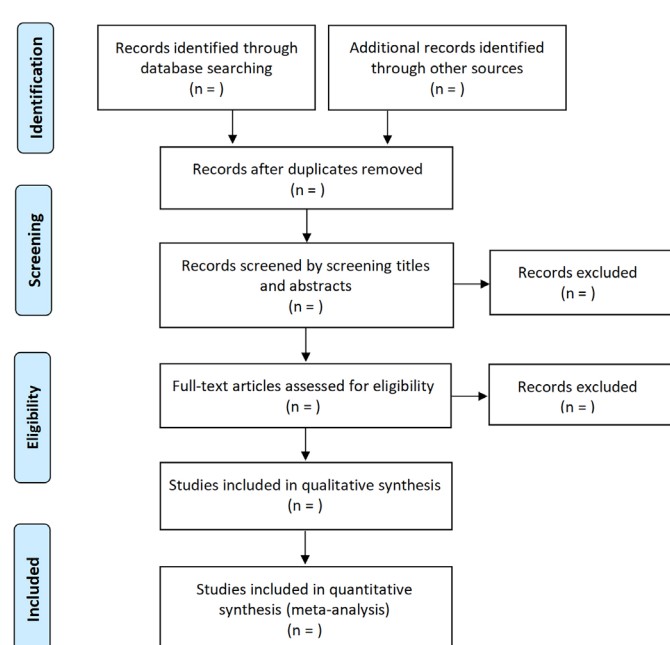

**Figure 1** Template of the Preferred Reporting Items for Systematic Reviews and Meta-Analyses flow chart.

about the optimal dose of perineural dexamethasone for PNB.

## METHODS AND ANALYSIS
### Study registration
This systematic review has been registered with the International Prospective Register of Systematic Reviews (registration number: CRD42022385672) and is reported according to the Preferred Reporting Items for Systematic review and Meta-Analysis Protocols guidance.

### Search strategy
Two authors will perform a comprehensive search in PubMed, EMBASE, the Cochrane Central Register of Controlled Trials and Web of Science from inception to 1 March 2023 to identify RCTs comparing the analgesic effects of various doses of perineural dexamethasone on PNB. Language will be restricted to English. The key terms used in our systematic search will include "Dexamethasone", "Peripheral nerve block" and "Randomized controlled trials". We will also track additional studies by reviewing the references of relative studies. The search strategy for PubMed is shown in table 1, and the search strategies for other databases are presented in online supplemental file 1.

### Inclusion and exclusion
Inclusion criteria: (a) study design: RCTs, (b) participants: adult patients (>18 years) with PNB, (c) comparisons: different doses of perineural dexamethasone as an adjuvant to local anaesthetics, and (d) primary outcome: duration of analgesia; secondary outcomes: pain scores, total analgesic requirement over 48 hours and adverse effects (eg, neuropraxia, nausea, vomiting, drowsiness, dizziness and constipation). Retrospective studies,

reviews, meta-analyses, case reports, conference abstracts, comments and studies regarding paediatric surgeries will be excluded.

### Study selection
Two authors will independently identify the eligible studies by screening the title and abstract, followed by reviewing the full texts of potentially eligible articles. Disagreements will be resolved by discussion with a third author. Study selection will be reported using a Preferred Reporting Items for Systematic Reviews and Meta-Analyses flow chart (figure 1).

### Data extraction
Two authors will independently extract the following information from included studies: authors, publication year, countries, sample size, patients' age, body mass index, the concentration of dexamethasone, types of surgery, types of anaesthesia and nerve block, interventions, controls, outcomes and postoperative analgesia. We will attempt to contact the corresponding author for raw data, if the results were reported by median with range. If there is no response, the median with range will be converted to mean with SD according to the methods described by Hozo et al.[24] Disagreements will be discussed with a third author.

### Risk of bias assessment
Two authors will independently assess the quality of included studies using the Cochrane Collaboration's tool for risk of bias.[25] The assessment for risk of bias contains six items: (a) random sequence generation (selection bias); (b) allocation concealment (selection bias); (c) blinding of participants and personnel (performance bias); (d) blinding of outcome assessment (detection

bias); (e) incomplete outcome data (attrition bias) and (f) selective reporting (reporting bias). The assessment for each item will be described as 'low', 'high' or 'unclear'. Disagreements will be discussed with a third author.

## Statistical analysis

The mean differences and risk ratios with 95% CIs will be calculated for continuous and dichotomous data, respectively. We will use the random-effects model to synthesise data due to the clinical heterogeneity (eg, type of surgery, anaesthesia and peripheral nerve block). Statistical heterogeneity will be assessed by the $I^2$ test. Significant heterogeneity will be considered when $I^2$ statistic is >50%. Subgroup analysis and meta-regression will be performed to explore the source of heterogeneity among studies. Sensitivity analyses will be conducted to assess whether the pooled results are robust. $P<0.05$ is considered statistically significant. The quality of evidence will be rated as 'high', 'moderate', 'low' or 'very low' using the Grading of Recommendation, Assessment, Development and Evaluation (GRADE) approach.

## Patient and public involvement

None.

## Ethics and dissemination

No ethical approval is required for this study. The results of this study will be submitted to peer-reviewed journals.

## DISCUSSION

Currently, various doses of perineural dexamethasone have been used in PNB to extend the duration of analgesia. Considering the potential neurotoxicity for perineural use of high dose of dexamethasone, it is meaningful to explore the optimal dose of perineural dexamethasone for PNB which will provide longer analgesia and lower incidence of adverse effects. This present protocol of a systematic review and meta-analysis will investigate whether there is dose dependency for perineural dexamethasone on the analgesic effects and explore its optimal dose for PNB. Therefore, the results of our study may provide guidelines for dose selection of perineural dexamethasone for PNB.

However, high clinical heterogeneity may exist due to several factors, such as the type and concentration of local anaesthetics, the type of surgery, anaesthesia and nerve blocks. Therefore, we will perform subgroup analysis and meta-regression to explore the heterogeneity resource. Moreover, sensitivity analyses will also be conducted to test whether the pooled results are robust. From the results of subgroup analysis, we may recommend the optimal dose of perineural dexamethasone for specific local anaesthetic in a specific type of PNB. Additionally, our results may facilitate more well-designed RCTs to be conducted to confirm the optimal dose of perineural dexamethasone for PNB.

**Contributors** Conceptualisation—WC and DZ. Data curation—SW. Formal analysis—WC. Methodology—WC and SW. Validation—WC and SW. Writing (original draft)—WC and DZ. Writing (review and editing)—WC.

**Funding** This study is supported by the Natural Science Foundation of Sichuan Province (grant no. 2022NSFSC1399 (to DZ)).

**Competing interests** None declared.

**Patient and public involvement** Patients and/or the public were not involved in the design, or conduct, or reporting, or dissemination plans of this research.

**Patient consent for publication** Not required.

**Provenance and peer review** Not commissioned; externally peer reviewed.

**ORCID iD**
Wenjie Chai http://orcid.org/0000-0003-0356-1148

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
