## [Reviewer comments · BMJ Open]

ARTICLE DETAILS

TITLE (PROVISIONAL)	Optimal dose of perineural dexamethasone for the prolongation of analgesia for peripheral nerve blocks: protocol for a systematic review and meta-analysis
AUTHORS	Chai, Wenjie; Wang, Shaopeng; Zhang, Donghang

VERSION 1 – REVIEW

REVIEWER	Kamiya, Yoshinori Gifu Daigaku Igakubu Daigakuin Igakukei Kenkyuka, Anesthesiology and Pain Medicine
REVIEW RETURNED	16-Apr-2023

GENERAL COMMENTS	I read this protocol manuscript, and I searched resemble meta-analysis in PubMed. I found several resembled manuscripts in PubMed, and I think this protocol would not provide new findings. Primarily, in this manuscript, the authors did not mention the meta-analysis performed by E Albrecht (Anaesthesia. 2015 Jan;70(1):71-83), and I think it is unfair. However, the report by E Albrecht was written in 2015, and new evidence should be presented. This protocol may show the latest evidence. Protocol itself is by proper method.
--

REVIEWER	Kim, David Hospital for Special Surgery, Weill Medical College of Cornell University, Department of Anaesthesiology, Critical Care and Pain Management
REVIEW RETURNED	25-Apr-2023

GENERAL COMMENTS	Commend Author on investigating the optimal dose of dexamethasone via literature search. However, the outcomes do not address the "safety" of perineural dexamethasone. There is no mention of Neuropraxia.
---

REVIEWER	Tanavalee, Chotetawan Chulalongkorn University, Department of Orthopaedics
REVIEW RETURNED	06-Jun-2023

GENERAL COMMENTS	I recommend including data such as BMI and the concentration of dexamethasone if it is feasible, as your focus is on dose and safety. BMI can affect the distribution of dexamethasone, and I believe that the concentration may be toxic to the peripheral nerves. This study could generate more interesting new points.
---

VERSION 1 – AUTHOR RESPONSE

Reviewer: 1

Dr. Yoshinori Kamiya, Gifu Daigaku Igakubu Daigakuin Igakukei Kenkyuka

Comments to the Author:

I read this protocol manuscript, and I searched resemble meta-analysis in PubMed. I found several resembled manuscripts in PubMed, and I think this protocol would not provide new findings. Primarily, in this manuscript, the authors did not mention the meta-analysis performed by E Albrecht (Anaesthesia. 2015 Jan;70(1):71-83), and I think it is unfair. However, the report by E Albrecht was written in 2015, and new evidence should be presented. This protocol may show the latest evidence.

Protocol itself is by proper method.

Response: Thank you very much for these helpful comments! We have cited and discussed the relevant previous meta-analyses on same topics in the Introduction section (page 4, line 12-22). Our meta-analysis will include several new RCTs, thus will update the previous meta-analysis and provide the latest evidence for the optimal dose of perineural dexamethasone for PNB.

Reviewer: 2

Dr. David Kim, Hospital for Special Surgery, Weill Medical College of Cornell University

Comments to the Author:

Commend Author on investigating the optimal dose of dexamethasone via literature search. However, the outcomes do not address the "safety" of perineural dexamethasone. There is no mention of Neuropraxia.

Response: Thank you very much for these positive comments! We have added 'Neuropraxia' as a secondary outcome to evaluate the safety of perineural dexamethasone (page 5, line 21).

Reviewer: 3

Dr. Chotetawan Tanavalee, Chulalongkorn University

Comments to the Author:

I recommend including data such as BMI and the concentration of dexamethasone if it is feasible, as your focus is on dose and safety. BMI can affect the distribution of dexamethasone, and I believe that the concentration may be toxic to the peripheral nerves. This study could generate more interesting new points.

Response: Thank you very much for these helpful suggestions! We agree with the reviewer that BMI can affect the distribution of dexamethasone, and the concentration may be toxic to the peripheral nerves. During this revision, we have added the information about BMI and the concentration of dexamethasone in 'Data extraction' section (page 6, line 1), which may generate more interesting new points.

VERSION 2 – REVIEW

REVIEWER	Kim, David Hospital for Special Surgery, Weill Medical College of Cornell University, Department of Anaesthesiology, Critical Care and Pain Management
REVIEW RETURNED	05-Jul-2023

GENERAL COMMENTS	Thanks for including neuropraxia as one of the secondary outcomes.
--

REVIEWER	Tanavalee, Chotetawan Chulalongkorn University, Department of Orthopaedics
REVIEW RETURNED	06-Jul-2023

GENERAL COMMENTS	Ok, accept the protocol.
--------------------------